

# Habitat ephemerality affects the evolution of contrasting growth strategies and cannibalism in anuran larvae

Dogeun Oh[*], Yongsu Kim[*], Sohee Yoo and Changku Kang

Department of Biosciences, Mokpo National University, Muan, Jeollanamdo, South Korea
[*] These authors contributed equally to this work.

## ABSTRACT

Ephemeral streams are challenging environments for tadpoles; thus, adaptive features that increase the survival of these larvae should be favored by natural selection. In this study, we compared the adaptive growth strategies of *Bombina orientalis* (the oriental fire-bellied toad) tadpoles from ephemeral streams with those of such tadpoles from non-ephemeral streams. Using a common garden experiment, we tested the interactive effects of location (ephemeral *vs.* non-ephemeral), food availability, and growing density on larval period, weight at metamorphosis, and cannibalism. We found that tadpoles from ephemeral streams underwent a shorter larval period compared with those from non-ephemeral streams but that this difference was contingent on food availability. The observed faster growth is likely to be an adaptive response because tadpoles in ephemeral streams experience more biotic/abiotic stressors, such as desiccation risk and limited resources, compared with those in non-ephemeral streams, with their earlier metamorphosis potentially resulting in survival benefits. As a trade-off for their faster growth, tadpoles from ephemeral streams generally had a lower body weight at metamorphosis compared with those from non-ephemeral streams. We also found lower cannibalism rates among tadpoles from ephemeral streams, which can be attributed to the indirect fitness costs of cannibalizing their kin. Our study demonstrates how ephemeral habitats have affected the evolutionary change in cannibalistic behaviors in anurans and provides additional evidence that natural selection has mediated the evolution of growth strategies of tadpoles in ephemeral streams.

## INTRODUCTION

Ephemeral streams and ponds are highly heterogeneous in various biotic and abiotic stressors, including hydroperiod, predation/competition, and food availability, thus an appropriate choice of where to stay heavily influences the survivorship of both individuals and their progeny (*Spieler & Linsenmair, 1997*; *Blaustein, 1999*; *Dayton & Fitzgerald, 2001*; *Stubbington et al., 2017*). Studies of ephemeral habitats have revealed the importance of parental care because females' choice of oviposition sites directly determines the fitness of their progeny (*Rieger, Binckley & Resetarits Jr, 2004*; *Pintar & Resetarits Jr, 2017a*). Accumulating evidence has consistently demonstrated the adaptive behavior of female

Corresponding author
Changku Kang,
changku.kang@gmail.com,
changkukang@mokpo.ac.kr

anurans: species in ephemeral habitats generally avoid sites that have a high risk of desiccation, predation, and competition (*Rudolf & Rödel, 2005*; *Buxton & Sperry, 2017*; *Pintar & Resetarits Jr, 2017b*), although exceptional peculiar adaptations, such as feeding their eggs to tadpoles, have also been reported (*Poelman & Dicke, 2007*).

Considerable research has been devoted to identifying the adaptive response of tadpoles to external stressors (*Lind, Persbo & Johansson, 2008*). In general, tadpoles develop faster if they are exposed to greater risks (*e.g.*, predation and desiccation) and slower when resources are limited (*Newman, 1989*; *Nicieza, 2000*; *Richter-Boix, Tejedo & Rezende, 2011*). For example, studies have demonstrated that frogs under higher risk of desiccation develop faster than those under lower risk (*Richter-Boix, Llorente & Montori, 2006*; *Richter-Boix, Tejedo & Rezende, 2011*; *Gomez-Mestre, Kulkarni & Buchholz, 2013*; *Pujol-Buxó et al., 2016*). These adaptive responses largely account for developmental plasticity of tadpoles and would provide a survival advantage to those inhabit ephemeral habitats (*Tejedo & Reques, 1994*; *Van Buskirk, 2002*; *Székely et al., 2017*; *Lent & Babbitt, 2020*). However, to date, evidence of how ephemeral habitats affect the degree of developmental plasticity and whether the habitat differences have shaped the genetic basis of anuran growth strategies is scarce.

Here, we studied the oriental fire-bellied toad (*Bombina orientalis*) to examine the patterns of evolution in shaping tadpole developmental strategies in ephemeral streams. *B. orientalis* is a semi-aquatic frog species distributed mainly across Eastern Asia. We primarily focused on comparing populations between ephemeral (Jeju Island) and non-ephemeral (mainland South Korea, hereinafter referred to as "mainland") streams. Importantly, unlike on the mainland, the surface of Jeju Island is covered by basalt layers formed by volcanic activities that drain water quickly into the underground. This geological feature has shaped most Jeju Island streams ephemeral, whereas the mainland's habitats consist of continuously flowing non-ephemeral streams (Fig. 1). *B. orientalis* colonized Jeju Island during the Cenozoic Era *via* a single dispersal event from the mainland (estimated mean population age, 4.32 Ma), and no gene flow has occurred since then (*Fong et al., 2016*). After this separation, the mainland and Jeju Island populations have diverged phenotypically (*Kang et al., 2017*).

The fundamental difference in habitat structure between the mainland and Jeju Island and the fact that gene flow has not been observed after the separation of populations provide an ideal testing ground for exploring how natural selection shapes the adaptive growth strategies of tadpoles in response to different environmental pressures. Because tadpoles in Jeju Island streams are isolated in small pools, with the surface area of most pools ranging from 0.01 to 36 m$^2$, during development at least until it rains (*Baek et al., 2021*), they confront several additional environmental stressors compared with tadpoles in mainland streams. First, they are exposed to a greater risk of desiccation such that even several days without rain can be fatal to those in pools with low hydroperiod. Second, resources are limited because movements among pools are impossible during the larval stage. Third, flooding occurs under heavy rain, which could potentially wash away all eggs and tadpoles downstream—where predatory risks are likely to increase owing to the presence of fish.

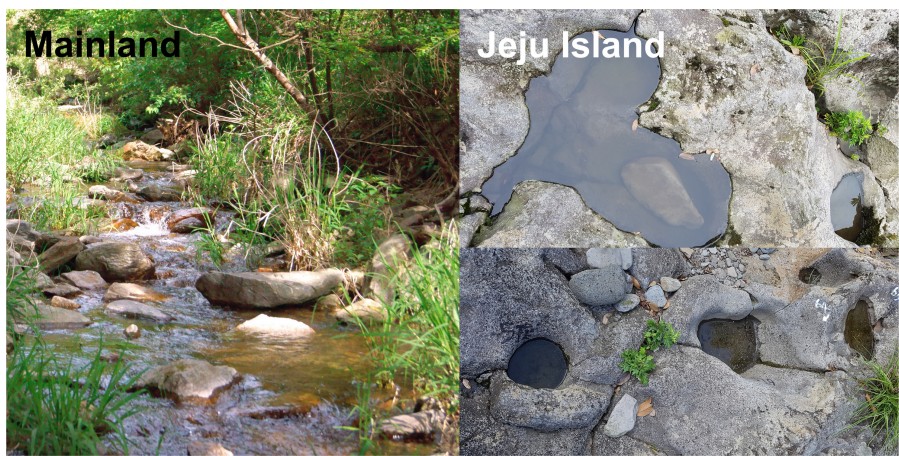

**Figure 1** Images showing the differences in habitat between mainland (left; non-ephemeral) and Jeju Island (right; ephemeral) streams.

Several predictions can be made regarding tadpole growth strategies based on the additional stressors imposed on ephemeral habitats. First, tadpoles in ephemeral streams would develop faster than those in non-ephemeral streams until they become froglets, when they are liberated from the environmental stressors associated with isolated pools (*Lind, Persbo & Johansson, 2008*; *Enriquez-Urzelai et al., 2013*; *Dittrich et al., 2016*). Second, as a trade-off for their faster growth, froglets in ephemeral streams would reach a smaller size at metamorphosis (*Edge, Thompson & Houlahan, 2013*). In addition, we predict that the two populations would differ in their cannibalistic behaviors. There are two alternative hypotheses about the degree of their cannibalism. First, because pools in ephemeral streams are isolated from one another, thus a closed system for organic matters, there is limited food available for tadpoles. Under this food-limited environment, natural selection may favor cannibalistic behaviors that increase individual survival and enhance development (*Crump, 1990*; *Wildy et al., 1998*). In this circumstance, tadpoles in ephemeral streams may be more cannibalistic than those in non-ephemeral streams (*Semlitsch & Reichling, 1989*; *Gould, Clulow & Clulow, 2020*). Alternatively, because *B. orientalis* females avoid laying their eggs in pools already occupied by conspecifics (*Baek et al., 2021*), tadpoles in the same pool are likely to be their kin; therefore, as cannibalizing their kin decreases tadpoles' indirect fitness (*Pfennig, 1997*; *Muralidhar et al., 2013*), natural selection may favor tadpoles in ephemeral streams that cannibalize less.

We tested the abovementioned predictions by comparing the growth of tadpoles from ephemeral streams with that of tadpoles from non-ephemeral streams using a common garden experiment. Moreover, to examine whether multiple factors interactively affect tadpole growth and cannibalistic behaviors, we manipulated two factors that are expected to differ between ephemeral and non-ephemeral habitats, namely, food availability and density (*Newman, 1987*; *Álvarez & Nicieza, 2002*; *Lent & Babbitt, 2020*).
## METHODS

### Experimental treatment and procedure

To examine whether differences in habitat structure between ephemeral and non-ephemeral stream populations have affected the evolution of their growth strategies, we conducted a common garden experiment using the progenies of frogs from the mainland and Jeju Island under water temperature control (22 °C). First, we collected adult frogs from streams in the mainland (34.7°–34.9°N, 126.5°–126.7°E) and Jeju Island (33.45°N, 126.56°E) and brought them to the laboratory. Each collection location is represented by a single population. We placed three to five individual frogs from both sexes together in a semi-aquatic terrarium (40 × 23 × 23 cm) filled with dechlorinated water, rocks, and aquatic plants to facilitate mating. We provided calcium and vitamin D–powered juvenile crickets and mealworms *ad libitum*. We maintained seven mating chambers for each location until we collected enough eggs to fill all experimental cages (as described below). In total, 67 Jeju Island frogs and 48 mainland frogs (including both sexes; approximate sex ratio, 1:1) contributed to egg-laying. We controlled the room temperature using an air conditioner (22 °C) and provided natural light through transparent windows.

Egg clutches were found every 1–4 days in all chambers. On the day that new egg clutches were found in at least three mating chambers, we collected the eggs and relocated them to experimental cages (25 × 16 × 17 cm) filled with 3.5 L of dechlorinated water where the eggs were allowed to develop. To provide genetic variability within and among experimental cages, we filled each cage with eggs from at least three mating chambers (a similar number of eggs from each mating chamber was used to fill each cage). We never filled the same treatment on the same day. The order of egg filling was random among experimental treatments. Egg collection was completed within two weeks (from 1st to 13th July 2020). Because (1) females do not typically lay eggs in consecutive days, and (2) all experimental chambers were filled in two weeks, we consider that the eggs used for the experiment were not from a limited number of frogs but many different families. We released all adult frogs back to the collection sites after then.

We manipulated three treatments in a full factorial design: (1) the density of tadpoles (low *vs.* high), (2) the amount of food provided each day (scarce *vs.* abundant), and (3) the location where parental frogs were collected (ephemeral *vs.* non-ephemeral streams). For density treatment, we kept either five (low-density) or 30 (high-density) individuals in each cage. For food treatment, we provided either 0.01 (scarce) or 0.1 g (abundant) of fish food (TetraBits Complete; Tetra, Germany) every day. We proportionally reduced the amount of food provided to cages in which some tadpoles had died or became froglets. We replaced half of the water every 2–3 days and removed any remaining food daily. We replicated each treatment twice and tested 280 tadpoles in total.

We surveyed the survival of all tadpoles and whether any tadpoles became froglets (Gosner stage 46) (*Gosner, 1960*) each day. When a tadpole reached Gosner stage 46, we calculated the number of days that had passed from oviposition (larval period), measured the weight of the froglet (weight at metamorphosis), and then brought it out of the cage. We considered tadpoles to have been cannibalized when either tadpoles were missing

or we directly observed cannibalistic behaviors. We note here that though the survival of tadpoles had been monitored a few times a day, we cannot distinguish between pre- and post-mortem cannibalism. We also noted the occurrence of non-cannibalistic deaths where tadpoles were found dead without any signs of attack. All froglets were released to the location where their parental frogs were collected at the end of the experiment. All protocols were approved by Mokpo National University Animal Care and Use Committee (approval no: MNU-IACUC-2020-001).

## Data analysis

We used R 4.0.2 (*R Core Team, 2018*) for all analyses. To deal with pseudo-replications within replicates (tadpoles in the same cages), we used a multi-level modelling approach and set each cage as a random factor in all analyses.

To compare the proportion of cannibalized tadpoles among treatments, we fitted generalized linear mixed models (GLMMs) with a binomial error distribution. We used a binary variable of whether each tadpole was cannibalized or not as a response variable and our treatments (food, density, and location) as predictors of up to two-way interactions. We also used GLMMs to compare the occurrence of non-cannibalistic deaths among treatments. For the comparisons of larval period and weight at metamorphosis, we used food, density, location, and their interactions as predictor variables with larval period and weight at metamorphosis as response variables. Larval period was log-transformed to mitigate right-skewness of the distribution. We also estimated whether the relationship between larval period and weight at metamorphosis differed among our treatments. For this, we fitted a linear regression using weight at metamorphosis as a response, larval period as a predictor for each cage. Then we extracted the slope of the fitted line and compared the slopes among our treatment groups. We first fitted the model with all predictors and sequentially removed non-significant terms based on Akaike Information Criterion (AIC). For the terms that were removed during this model selection, we presented the statistics when the term was removed from the model. We used the Wald test to find out the parameter of each explanatory variable is zero.

## RESULTS

### Comparison of non-cannibalistic deaths and cannibalism

Overall, 151 tadpoles (54% of the total) that survived until they became froglets were measured for larval period and weight. Of the tadpoles that had died, 40 (14%) did not show any signs of cannibalism and 89 (32%) were considered to have been cannibalized. Low food availability primarily affected the occurrence of non-cannibalistic deaths ($z = 2.35$, $P = 0.02$). The number of individuals found dead under the scarce food and high-density condition comprised 65% of all non-cannibalistic deaths. No other predictor explained the occurrence of non-cannibalistic deaths (all $P > 0.15$).

Putative cannibalism occurred more frequently when food was scarce ($z = -2.03$, $P = 0.04$) and among mainland tadpoles ($z = -3.37$, $P < 0.001$). The density effect showed border-line evidence ($z = -1.69$, $P = 0.06$). Although the interactions between all predictors were not statistically significant (all $P > 0.15$), we further explored their

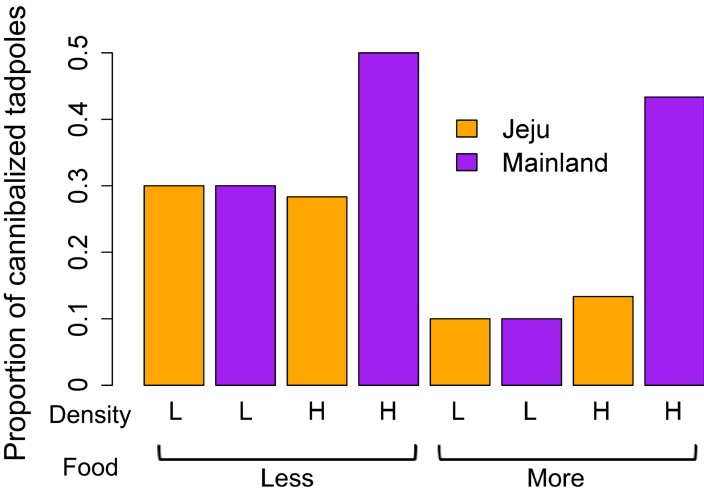

**Figure 2** Comparison of the mean proportion of cannibalized tadpoles among treatments (density, food, and location).

interactive effects between density and location variables owing to the noticeable patterns that emerged in the graph (Fig. 2). Individual analysis of the density groups revealed that food ($z = -1.52$, $P = 0.12$), location ($z = 0$, $P = 1$), and the interaction between these predictors ($z = 0$, $P = 1$) had no effect on the occurrence of cannibalism in the low-density groups. However, tadpoles from ephemeral streams cannibalized less than did those from non-ephemeral streams in the high-density groups ($z = -4.17$, $P < 0.001$). In the high-density group, we also found borderline evidence that cannibalism occurred more frequently when food was scarce ($z = -1.84$, $P = 0.06$) and determined that the interaction between location and food treatments had no effect ($z = 1.12$, $P = 0.26$). We consider that the interactive effect between density and location in the main analysis was obscured due to no location effect in low-density groups despite a significant difference in cannibalism rates in high-density groups.

## Comparison of larval period and weight at metamorphosis

Analysis of the larval period showed that our treatments interactively affected the larval period of tadpoles (Fig. 3A; see Table 1 for full statistics). First, the effect of location depended on food levels: tadpoles from ephemeral streams had shorter larval periods than those from non-ephemeral streams when food was abundant, while we found no consistent effect of location in abundant food conditions (Fig. 3A). More specifically, tadpoles from ephemeral streams became froglets 3–4.5 days (6%–10%) faster on average compared with those from non-ephemeral streams under abundant food and scarce-food × low-density conditions. However, tadpoles from ephemeral streams became froglets 12.6 days (17%) later on average than those from non-ephemeral streams under scarce-food × high-density

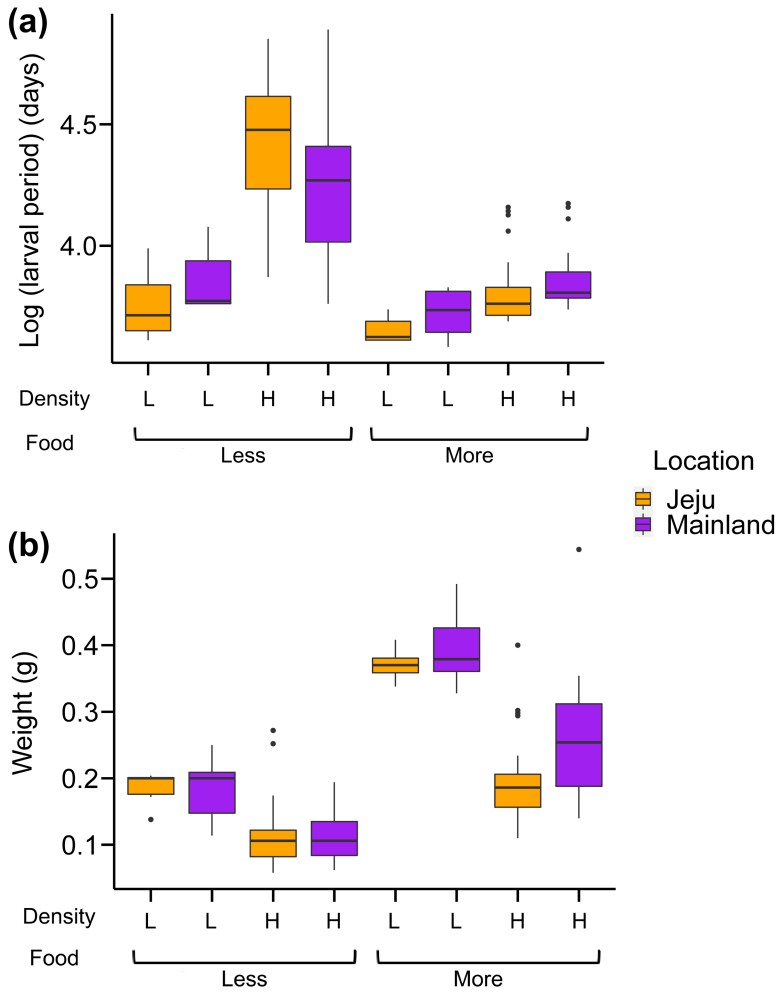

**Figure 3** Box-whisker plots showing the effects of density, food, and location on (A) larval period (days passed from oviposition to Gosner stage 46) and (B) weight at metamorphosis.

conditions. The larval periods under this harsh condition were substantially longer than those in other treatments (Fig. 3A).

The weights at metamorphosis were generally heavier under the abundant-food and low-density conditions (Fig. 3B; Table 1). Froglets from ephemeral streams were generally lighter than those from non-ephemeral streams. The interaction effect was only significant between food and density treatments: weights at metamorphosis were consistently heavier under low-density conditions than high-density conditions, but this difference was more prominent in abundant-food conditions than scarce-food conditions. We also note here that the effect of location on weights at metamorphosis was principally driven by the weight difference in abundant-food conditions (Fig. 3B), with the $P$-values of the interaction between location and food was marginally significant ($P = 0.08$).

**Table 1 Summary of linear mixed models (LMMs) examining the effects of food, density, and location on larval period and weight at metamorphosis.**

| Response | Predictor | Coefficient (s.e.) | Wald t | P |
|---|---|---|---|---|
| Larval period | Food[a] | 0.20 (0.08) | 2.57 | 0.01 |
| | Density[b] | 0.14 (0.05) | 2.65 | 0.009 |
| | Location[c] | 0.06 (0.04) | 1.57 | 0.12 |
| | Food × Density | 0.41 (0.08) | 5.13 | <0.001 |
| | Food × Location | −0.18 (0.07) | −2.74 | 0.007 |
| | Density × Location | −0.12 (0.08) | −1.55 | 0.12 |
| Weight at metamorphosis | Food[a] | −0.17 (0.03) | −6.27 | <0.001 |
| | Density[b] | −0.16 (0.02) | −8.22 | <0.001 |
| | Location[c] | 0.05 (0.02) | 2.95 | <0.001 |
| | Food × Density | 0.09 (0.03) | 3.12 | 0.02 |
| | Food × Location | −0.05 (0.03) | −1.95 | 0.08 |
| | Density × Location | 0.02 (0.03) | 0.83 | 0.42 |

**Notes.**
[a] Scarce *vs.* abundant.
[b] Low *vs.* high.
[c] Jeju Island (ephemeral stream) *vs.* mainland South Korea (non-ephemeral stream).

The relationship between weight at metamorphosis and larval period (*i.e.,* the slope from linear regressions between the two variables) did not differ among our treatment groups (all $P > 0.12$; see Fig. S1).

# DISCUSSION

Our results demonstrate three major differences between the mainland (non-ephemeral habitat) and Jeju Island (ephemeral habitat) tadpoles. First, the larval period was generally shorter for Jeju Island tadpoles, although this difference was conditional. Second, there was substantial variation in weight at metamorphosis among the treatments largely induced by the correlation between larval period and weight at metamorphosis (Fig. S1). However, the difference between the Jeju Island and mainland populations was only apparent when food was abundant. Third, Jeju Island tadpoles cannibalized less than did mainland tadpoles.

Our results align well with the prediction that tadpoles in more variable and unpredictable environments would grow faster than those in more stable environments: tadpoles from Jeju Island grew faster than did those from the mainland in all comparisons except when both available resources were scarce and density was high, thereby intensifying the competition for food. Early metamorphosis is not always preferred because it often results in a smaller size at metamorphosis, which can affect post-metamorphic growth and survival (*Altwegg & Reyer, 2003*; *Edge, Thompson & Houlahan, 2013*; *Skékely et al., 2020*). Therefore, early metamorphosis should evolve only when the benefits of developing earlier exceed the costs of having a smaller size. The survival advantage seems to be a key factor that has facilitated the observed earlier metamorphosis of Jeju Island tadpoles because early metamorphosis would benefit tadpoles living under higher environmental stressors more. The larval period was substantially extended under the food-limited and high-density conditions for both tadpole populations (on average, a 37-day extension, which corresponds

to an 83% increase in the larval period), suggesting that adaptive responses were hampered under this harsh condition. We also found a general trend that weight at metamorphosis is positively correlated with the larval period within each treatment (Fig. S1). However, a comparison of the two populations revealed that this correlation was somewhat alleviated in Jeju Island tadpoles: weights at metamorphosis were heavier in mainland tadpoles when there was enough food, whereas weights were similar when food was limited.

Collectively, the Jeju Island tadpoles generally grew faster than did the mainland tadpoles with a sacrifice in their weights at metamorphosis. Tadpoles from both regions showed high degree of developmental plasticity, suggesting that plastic response can be adaptive in both regions. However, the degree of plasticity differed between the two regions. The standard deviation of larval period was larger for Jeju Island tadpoles (s.d. = 24.13; range from 37 to 128 days) than mainland frogs (s.d. = 19.73; range from 36 to 133 days). In contrast, the standard deviation of weight at metamorphosis was larger for mainland frogs (s.d. = 0.11; range from 0.062 to 0.544 g) than Jeju Island frogs (s.d. = 0.08; range from 0.058 to 0.408) primarily because the upper weight limit was lower in Jeju Island tadpoles than mainland frogs. Thus, the difference in plastic response at least partly explains the geographical differences. In addition, because Jeju Island frogs have been separated from mainland frogs since their initial colonization (*Fong et al., 2016*), the genetic differences between them should also account for the observed differences. One (or another) way that may induce the evolution of different growth strategies is through different maternal investments. It has been demonstrated that the initial maternal invertment (often represented by egg size) affects larval period, weight at metamorphosis, and cannibalistic behaviors (*Kaplan, 1992*; *Laugen et al., 2005*; *Martin & Pfennig, 2010*). Although we did not measure egg size, Jeju Island frogs may allocate less resource to each egg than mainland frogs because there exist more external stressors that threat tadpole survival in their habitat (ephemeral streams), thus investing more in one egg can be risky. Whether Jeju Island and mainland frogs have different maternal strategies remains open for future studies.

The observed difference between mainland and Jeju Island tadpoles may not only account for habitat differences, but other geographical or insular effects could also have affected given that our parental frogs were collected from one population in each location. Although we collected parental frogs from the narrow geographical area in both the mainland and Jeju Island, there are not many geographical barriers preventing gene flow within both locations (*Fong et al., 2016*). Thus, we consider that the geographical effects within each location should be inconsequential to the observed mainland-island difference. Disentangling these effects are challenging, but multiple populations and community-wise comparisons (comparisons of multiples species sharing the same habitats) could reveal the importance of habitat features in shaping tadpole growth strategies.

Previous studies show that cannibalism occurs more frequently under food-limited and high-density conditions (*Wildy et al., 2001*; *Vaissi & Sharifi, 2016*). Our results seem to be congruent with these findings, though we cannot discriminate between pre- and post-mortem cannibalism. The difference between Jeju Island and mainland tadpoles was prominent only when the growing density was high, thus providing more opportunities for cannibalism. Although we cannot discriminate between pre- and post-mortem cannibalism,

cannibalism rates were consistently higher among mainland tadpoles even in abundant-food conditions where non-cannibal mortality was low. In addition, we have no reason to consider that mainland tadpoles happened to die more than Jeju Island tadpoles and scavenged more after then. This suggests that the higher cannibalism rates among mainland tadpoles were not just a by-product of scavenging dead individuals. Our finding that Jeju Island tadpoles cannibalized less than did mainland tadpoles can be explained by indirect fitness costs (*Pfennig, 1997*; *Schultner et al., 2014*). In ephemeral habitats, such as Jeju Island, that consist of a large number of small pools, tadpoles living in the same pool are likely to be their kin, especially when the parental frogs are known to avoid conspecific cues, as was previously demonstrated in *B. orientalis* Jeju Island populations ( *Baek et al., 2021*). Thus, the fitness advantage of cannibalizing conspecifics should be lower in Jeju Island tadpoles because of the indirect fitness costs of cannibalizing their kin. Lower cannibalism rates in Jeju Island tadpoles support this kin selection view.

In conclusion, our study highlights that environmental differences  have shaped the divergence in growth strategies between tadpoles from ephemeral streams and those from non-ephemeral streams. Differences in growth strategies generally follow predictions of adaptation to ephemeral conditions (*Richter-Boix, Llorente & Montori, 2006*; *Dittrich et al., 2016*), but we found that this effect was contingent on growing density and food availability. We also found lower cannibalism rates among tadpoles from ephemeral streams, which may be accounted for by the indirect fitness costs involved (*Pfennig, 1997*; *Dugas et al., 2016*). By comparing two geographically isolated populations using a common garden experiment, we have confirmed that the observed differences at least partly have a genetic basis. Environmental differences between mainland South Korea and Jeju Island provide a unique opportunity to explore how natural selection shapes phenotypic divergence in anurans through local adaptation. We also speculate that differences in tadpoles' developmental/physiological responses to other factors such as temperature and hydroperiod, exist, but this hypothesis remains to be tested.

## ACKNOWLEDGEMENTS

We thank C Park, S Noh, Y Hwang, and W Lim for their invaluable help in collecting frogs and caring for the tadpoles used in this study.

### Funding

This work was supported by the National Research Foundation of Korea (Grant No. 2019R1C1C1002466). The funders had no role in study design, data collection and analysis, decision to publish, or preparation of the manuscript.

### Grant Disclosures

The following grant information was disclosed by the authors:
National Research Foundation of Korea: 2019R1C1C1002466.

## Competing Interests
The authors declare there are no competing interests.

## Author Contributions

- Dogeun Oh and Yongsu Kim performed the experiments, analyzed the data, prepared figures and/or tables, authored or reviewed drafts of the paper, and approved the final draft.
- Sohee Yoo performed the experiments, authored or reviewed drafts of the paper, and approved the final draft.
- Changku Kang conceived and designed the experiments, analyzed the data, prepared figures and/or tables, authored or reviewed drafts of the paper, and approved the final draft.

## Animal Ethics
The following information was supplied relating to ethical approvals (i.e., approving body and any reference numbers):

All protocols were approved by the institutional committee for animal care and use (approval no: MNU-IACUC-2020-001).

## Data Availability
The raw data is available in the Supplementary File.

## Supplemental Information
Supplemental information for this article can be found online at http://dx.doi.org/10.7717/peerj.12172#supplemental-information.

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
