# Peer review of "Habitat ephemerality affects the evolution of contrasting growth strategies and cannibalism in anuran larvae"

_PeerJ, doi:10.7717/peerj.12172_

## Round 0.1 · original submission · Major Revisions

Please revise to address the reviewers’ concerns.

Reviewer 1 ·

Basic reporting

BASIC REPORTING
• Clear, unambiguous, professional English language used throughout.
o Yes, for the most part. See Specific Comments below for exceptions.
• Intro & background to show context.
o Yes, though some additional clarification would help. See Specific Comments below for exceptions.
• Literature well referenced & relevant.
o Yes, for the most part. See Specific Comments below for exceptions.
• Structure conforms to PeerJ standards, discipline norm, or improved for clarity.
o Yes, looks fine.
• Figures are relevant, high quality, well-labelled & described.
o Yes, although I think they could do with some additional detail in both the figures and the captions. See Specific Comments below.
• Raw data supplied (see PeerJ policy).
o Yes. Can the authors please explain the NAs in the data? Are they tadpoels that died during the experiment? How were the NAs dealt with in the analyses?

Vertebrate animal usage checks
• Have you checked the authors ethical approval statement?
o Yes, looks fine.
• Were the experiments necessary and ethical?
o Yes, no problems.
• Have you checked our animal research policies?
o Yes, looks fine.

Experimental design

EXPERIMENTAL DESIGN
• Original primary research within Scope of the journal.
• Research question well defined, relevant & meaningful. It is stated how the research fills an identified knowledge gap.
o Somewhat. See Specific Comments below for exceptions, especially regarding the Introduction.
• Rigorous investigation performed to a high technical & ethical standard.
o For the most part. I have concerns regarding genetic diversity in the experiment and the inclusion of only a single population from each location. See Specific Comments below.
• Methods described with sufficient detail & information to replicate.
o For the most part. The descriptions of statistical analyses, in particular, are insufficiently detailed. See Specific Comments below.

Validity of the findings

VALIDITY OF THE FINDINGS
• Impact and novelty not assessed.
• Meaningful replication encouraged where rationale & benefit to literature is clearly stated.
o Yes. It is especially important that this study be repeated with additional populations and that it be complimented by population-genetic studies to investigate some of the authors’ claims. See Specific Comments below.
• All underlying data have been provided; they are robust, statistically sound, & controlled.
o For the most part. See comment above regarding NAs. Also, considering habitat ephemerality is highlighted as a likely important selective pressure, I am surprised by the lack of a desiccation-risk treatment and accompanying control, as has been done with other taxa in studies by, for example, Denver, Gomez-Maestre, Buchholz, and others.
• Speculation is welcome, but should be identified as such.
o Some statements are made too strongly and presented somewhat axiomatically as explanations for the observed results despite a lack of data to support those claims. See Specific Comments below, especially regarding the Discussion.
• Conclusions are well stated, linked to original research question, & limited to supporting results.
o Somewhat, though some results are not fully explored (e.g., the big difference between insular and mainland in the abundant-food low-density treatment). In addition, conclusions regarding the potential for genetic differences to underpin the observed phenotypic differences, as well as a couple of other statements, seem to me to exceed the boundaries of the data without explicitly mentioning the speculative nature of those statements or alternative explanations. See Specific Comments below.

Additional comments

Specific Comments

Introduction
• line 38 – this sentence is vague and doesn’t communicate much of anything. Consider something that is more direct (e.g., “ephemeral environments impose numerous selective pressures and have favored the evolution of...”)
• line 40 – replace the slash with ‘and’
• line 42 – add a comma between ‘stay’ and ‘heavily’
• line 50 – adaptive responses to what?
• lines 57-62 – can you clarify these statements? Are you saying that prior studies that examined differences in tadpole development time produced results that were confounded with phylogenetic relatedness or other variables? I would suggest that past work by Denver, Gomez-Maestre, Buchholz, and others has effectively isolated the effects of desiccation risk on development time. Also, the final sentence is confusing in the context of the rest of the paragraph, which talks about plasticity, which should evolve in reliably predictable but variable environments, but the final sentence seems to ask whether directional selection might favor the evolution of singular developmental strategies. Clarifying this paragraph will help explain the value of your study.

Methods
• line 129-131 – were you able to track which matings the tadpoles in each replicate came from? If not, can you account for the possibility that your results might reflect, in part, outsized contributions of one or more families within treatments or replicates?
• line 132-133 – did you measure the parents’ SVL and/or mass? Adult body size and body condition are known to affect tadpole size, development, morphology, etc. in multiple anuran taxa (see, e.g., Bennett, A.M. and Murray, D.L., 2014. Maternal body condition influences magnitude of anti-predator response in offspring. Proceedings of the Royal Society B: Biological Sciences, 281(1794), p.20141806.
• line 161-162 – what distribution was specified for these models?
• line 166-168 – I see you used AIC scores for model comparisons, but where did your z-score test stats some from? Wald tests? Please include these details.
• An additional analysis that might improve your study would be a comparison of the relationship between larval period and weight at metamorphosis between island and mainland populations and between treatments (i.e., do the slopes differ?). This could generate an informative figure(s) if there are different slopes between locations/treatments. I see that you did part of this with Fig. S1, but there don’t seem to be any accompanying analyses.

Results
• line 175 – the z-score is reported with a comma where there should be a period
• line 183 – there is no Fig. 2B. Is this a typo?
• line 194 and Table 1 – please explain the statistical tests used here. I see t-statistics but no explanation of the means used to approximate degrees of freedom for these t-tests. Did you use a specific R package to perform these (e.g., lmerTest)? If so, please cite the package(s). Also, please include SE for at least the model coefficients.
• line 197 – ‘becoming’ should be ‘became’
• General – can you perform post-hoc tests to report pairwise differences among the treatment/location groups with corrected p-values in the results shown in Figs. 2 & 3? You’ve shown only whether treatments had effects in general.
• Figures
o can you use those posthoc tests to add indications to your plots of which groups are significantly different from other groups? (E.g., line over pairs of boxplots with p-values or asterisks that are explained in the captions)
o Please describe the box-and-whisker plots in the figure captions (e.g., thick middle line is median, boxes are interquartile ranges, whiskers are 1.5x IQR, points are individuals with values outside 1.5x IQR...s)

Discussion
• line 223-227 – see comment above regarding analysis of and potential figure(s) for the possibility different relationships between larval period and weight at metamorphosis among groups. I think it would be nice to see this addressed more explicitly.
• line 233-234 – I see positive correlations in all panels except the mainland abundant-food low-density one. Also, the caption for Fig. S1 contradicts the statement at these lines.
• line 239 – 243 – I would certainly expect genetic differences given the split in the Cenozoic and the subsequent lack of gene flow, but you do not present data that show genetic differences that are the bases of the phenotypic differences that you show in your study. Since anuran tadpoles are known to be highly plastic along various phenotypic axes, it seems a good idea to mention the possibility of plasticity playing some role in the observed differences. Also, is the insular habitat genuinely “unpredictable” as suggested on line 220? Or might there be reliable cues in the island habitat that could have favored the evolution of greater plasticity among island tadpoles?
• line 246 – is the fact that each location is represented by a single population stated clearly in the methods?
• line 247-250 – have you or anyone else investigated the populations genetics of the mainland and insular populations? how reasonable is it to assume homogeneity with each location?
• line 253 – please clarify how it might ‘drive’ faster larval growth
• line 257 – I think the indirect fitness/kin selection hypothesis is a bit overstated here, especially since you cannot distinguish between scavenging and cannibalism that actually consists of consuming live kin. It may be that the mainland tadpoles simply didn’t do as well in these experimental conditions and were quickly eaten after dying of other causes. Can you perform an analysis using only confirmed cannibalism of live kin, or are there not sufficient data?
• line 278-279 – I think this is stated too strongly. It is still possible that some portion of the observed differences arose via plasticity. Additional study would be required to demonstrate the point made in these lines.
• General – can you perhaps provide further hypotheses for the dramatic differences between insular and mainland populations’ weight-larval period relationships in the abundant-food low-density treatment (Fig. S1)?

References
• please refer to the PeerJ references instructions: https://peerj.com/about/author-instructions/#reference-format
• journal titles should be given in full and italicized
• please make sure other instructions are also followed

Reviewer 2 ·

Basic reporting

Language and clarity could be improved in a few places. Listed below are small edit suggestions.

Line 34: typo B. orientalis

Edit introduction first few sentences – for example:
Line 38: central to the [study of] environmental adaptations of animals?
Line 42: heavily influences
Line 42: some unclear sentence structure (who is “themselves”)?

Line 175: typo 2.35

Line 183: Fig. 2B should just be Fig. 2?

Line 197: became

Figure 1: I’m curious, are the streams on Jeju island disconnected puddles most of the time vs a flowing stream? How quickly does a stream go from flooded to puddles?

Fig S1: It may be better to have weight on the same scale between the 2 panels to allow for a direct comparison

Experimental design

This manuscript shows a well-conducted experiment evaluating whether there are evolved differences between a mainland and island population of Bombina orientalis, which differ in stream ephemerality. They test clear hypotheses that are well supported in the literature.

A few spots to clarify in methods:
Line 117: Clarify that these streams represent a single population – otherwise this is not clear until the discussion

Line 123-124: I’m unclear on your number of families – are you reporting all parents or the number of females that laid eggs?

Line 133: It could be helpful to provide a summarizing statement clarifying sample size in the experiment – how many different families are represented in the experiment? Are eggs from each family/clutch split between treatments or all in one experimental container?

Validity of the findings

Though only one mainland/non-ephemeral and one Jeju island/ephemeral population are used in the study, the authors are sufficiently careful not to over-reach the implications of their results.

Additional comments

The authors conclude that the observed differences have a genetic basis (i.e. lines 240, 280), yet since parents were collected directly from the field, environmental maternal/parental effects cannot be ruled out. Maternal effects can have a large effect on anuran phenotypes (see discussion of maternal effects in common garden experiments in Laugen et al 2005). For instance, in another cannibalistic tadpole, spadefoot toads, higher condition mothers produce larger eggs which become larger and more carnivorous tadpoles (Martin & Pfennig 2010). While it is impractical to rear the frogs for one or more generations under common conditions to minimize maternal effects, some studies measure egg size as a proxy for maternal investment and parental effects (i.e. Urszán et al 2015). I suggest the authors add a brief note or discussion acknowledging that maternal effects cannot be ruled out in this study.

Laugen, A. T., Kruuk, L. E., Laurila, A., Räsänen, K., Stone, J., & Merilä, J. (2005). Quantitative genetics of larval life-history traits in Rana temporaria in different environmental conditions. Genetics Research, 86(3), 161-170.

Martin, R. A., & Pfennig, D. W. (2010). Maternal investment influences expression of resource polymorphism in amphibians: implications for the evolution of novel resource-use phenotypes. PLoS One, 5(2), e9117.

Urszán, T. J., Török, J., Hettyey, A., Garamszegi, L. Z., & Herczeg, G. (2015). Behavioural consistency and life history of Rana dalmatina tadpoles. Oecologia, 178(1), 129-140.

·

Basic reporting

I very much enjoyed reading this manuscript. The system is appropriate to the questions, and the authors have done an excellent job explaining the natural history, and how it fits with these questions, to the reader. The work is well-designed and the conclusions are clear and interesting.

1) This is a well-organized and largely well-written paper. I do think it would benefit from just a bit more proofreading/language editing (although it is again well done). e.g., ln 196/7 I think there's a proofreading thing, and in several places some precise word choice will improve the manuscript (e.g., ln 238 replace "grew" with "developed"). This is a minor concern, but it is a great paper and I would like to see it have the biggest impact.

2) The literature review and use of references was good. It certainly isn't exhaustive, and there are some places where the classics might play a bigger role (e.g., citing some more broad kin selection papers rather than anuran empirical papers only), but again this is fairly minor. The authors covered the broad topics well.

3) The paper is professionally structured, and I have no concerns there. The raw data seem easy to access and interpret.

4) This work was self-contained, and had a complete and solid narrative structure.

Experimental design

1) This is well within the aims and scope of the journal

2) The research questions, both particular (104-109) and general (ln 60-62) were well-defined and addressed by the study and Discussion.

3) The work was well-conducted. It may have been nice to be able to keep track of families (e.g., with a random effect), but by mixing tadpoles from many families the authors have exceeded the standards in this field. I have no concerns about the quality of the work performed.

4) The methods were well-described. Indeed, the authors did an exceptional job of "telling the story".

Validity of the findings

The underlying data are provided and are straightforward and easy to interpret.

The statistics seem entirely appropriate, and the authors have done a nice job of including tables/figures for inspection and also telling a clear and and interesting (but not repetitive) story in text.

The conclusions seem safe to me, although I would encourage the authors to make a clear distinction between "splits" that are justified by significant interactions and ones that are not. If there's no interaction, there's no interaction, and follow up "differences" can be mentioned but should not be emphasized (ln 183).

Additional comments

The authors indicate fairly strongly that differences among habitats are likely genetic, and this is a theme throughout the paper (it even seems a prediction that is being tested).

Most conservatively, several generations should be bred in the lab when testing this hypothesis to remove environmental effects on parents that are transmitted as maternal effects. That is obviously impractical here. This makes no difference to the interest of the work, so I would simply re-write to suggest that differences may be genetic (or may not be) - because plasticity and genetic differences should push populations in the same direction in response to selective pressures, it genuinely makes no difference here.

I would also think a bit about whether adult body size differences matter. Does this put the differences in mass at metamorphosis in any different context? Maybe yes, maybe no. I think the potential interactions address this to some extent.

---

## Round 0.2 · accepted · Accept

Thank you for your conscientious attention to the reviewers' comments. I am happy to inform you that your paper has now been accepted for publication.

Reviewer 1 ·

Basic reporting

No comment.

Experimental design

No comment.

Validity of the findings

No comment.

Additional comments

I am satisfied that the authors addressed each point raised. I think this is a nice paper that is ready for acceptance and that the authors' data and presentation of their data will make for nice contributions to PeerJ and to the field.